# 11.4% Efficiency non-fullerene polymer solar cells with trialkylsilyl substituted 2D-conjugated polymer as donor

Haijun Bin[1,2,*], Liang Gao[1,2,*], Zhi-Guo Zhang[1], Yankang Yang[1,2], Yindong Zhang[3,4], Chunfeng Zhang[3,4], Shanshan Chen[5], Lingwei Xue[1], Changduk Yang[5], Min Xiao[3,4] & Yongfang Li[1,2,6]

Simutaneously high open circuit voltage and high short circuit current density is a big challenge for achieving high efficiency polymer solar cells due to the excitonic nature of organic semdonductors. Herein, we developed a trialkylsilyl substituted 2D-conjugated polymer with the highest occupied molecular orbital level down-shifted by Si–C bond inter-action. The polymer solar cells obtained by pairing this polymer with a non-fullerene acceptor demonstrated a high power conversion efficiency of 11.41% with both high open circuit voltage of 0.94 V and high short circuit current density of 17.32 mA cm$^{-2}$ benefitted from the complementary absorption of the donor and acceptor, and the high hole transfer efficiency from acceptor to donor although the highest occupied molecular orbital level difference between the donor and acceptor is only 0.11 eV. The results indicate that the alkylsilyl substitution is an effective way in designing high performance conjugated polymer photovoltaic materials.

[1] Beijing National Laboratory for Molecular Sciences, CAS Key Laboratory of Organic Solids, Institute of Chemistry, Chinese Academy of Sciences, Beijing 100190, China. [2] University of Chinese Academy of Sciences, Beijing 100049, China. [3] National Laboratory of Solid State Microstructures, School of Physics, and Collaborative Innovation Center of Advanced Microstructures, Nanjing University, Nanjing 210093, China. [4] Synergetic Innovation Center in Quantum Information and Quantum Physics, University of Science and Technology of China, Hefei, Anhui 230026, China. [5] Department of Energy Engineering, School of Energy and Chemical Engineering, Low Dimensional Carbon Materials Center, Ulsan National Institute of Science and Technology (UNIST), Ulsan 689-798, South Korea. [6] State and Local Joint Engineering Laboratory for Novel Functional Polymeric Materials, Laboratory of Advanced Optoelectronic Materials, College of Chemistry, Chemical Engineering and Materials Science, Soochow University, Suzhou, Jiangsu 215123, China. * These authors contribute equally to this work. Correspondence and requests for materials should be addressed to Z.-G.Z. (email: zgzhangwhu@iccas.ac.cn) or to Y.L. (email: liyf@iccas.ac.cn).

Bulk-heterojunction polymer solar cells (PSCs) are composed of a blend active layer of a *p*-type conjugated polymer as donor and an *n*-type semiconductor as acceptor sandwiched between an anode and a cathode where at least one of the two electrodes should be transparent[1–3]. The *n*-type semiconductors used as acceptor in the PSCs include fullerene derivatives[4], *n*-type conjugated polymers[5–12] and *n*-type organic semiconductors (*n*-OS)[13–23]. Among the acceptors, *n*-OS acceptors show distinguished advantages of easy tuning of absorption and electronic energy levels, strong absorbance, good morphology stability and more suitable for flexible devices. Therefore, the non-fullerene PSCs with *n*-OS as acceptor have attracted great attention recently, and their power conversion efficiency (PCE) has rapidly increased to 9–11% (refs 24–26).

In PSCs, due to the low dielectric permittivity and relatively large excitons binding energy of organic semiconductor, polymer donor and acceptor with cascading energy levels (the lowest unoccupied molecular orbital (LUMO) and highest occupied molecular orbital (HOMO) levels of the conjugated polymer donor should be higher than the corresponding LUMO and HOMO levels of the acceptor) are specially required to provide a driving force for excitons dissociation at the heterojunction interface[27–30]. In this cascade model, it is generally accepted that the LUMO/HOMO energy levels differences between the donor and acceptor ($\Delta E_{LUMO}$ and $\Delta E_{HOMO}$) should be larger than 0.3 eV for efficient excitons dissociation to overcome binding energy (usually 0.3–0.5 eV) of the excitons produced in the donor and acceptor phases by absorbing photons[27,30–32]. One crucial issue of the PSCs is their relatively large photon energy loss ($E_{loss}$) at 1 sun illumination which is defined as $E_{loss} = E_g - eV_{oc}$, where $E_g$ is the lowest optical bandgap of the donor and acceptor components[27,33–35]. The reported $E_{loss}$ in the most efficient fullerene-based PSCs is typically 0.7–1.0 eV. The larger $E_{loss}$ is associated with the larger driving force for excitons dissociation and the relative large non-radiative recombinations[27,28,36,37], thus creating a great challenge to simultaneously obtain a large open circuit voltage ($V_{oc}$) and a high short circuit current density ($J_{sc}$) in the PSCs.

In the non-fullerene PSCs, medium bandgap polymer donor–low bandgap n-OS acceptor pairs are particular interesting due to their complementary absorption in the visible-near infrared region, providing an effective approach for getting a high $J_{sc}$. In these systems, another interesting feature is the easy energy level matching to get higher $V_{oc}$ with lower $E_{loss}$ of 0.6–0.7 eV (refs 21,25,26,38). For example, the non-fullerene PSCs based on a medium bandgap D-A copolymer PffT2-FTAZ-2DT as donor and a low bandgap n-OS IEIC (ref. 19) ($E_g$ of 1.59 eV) as acceptor with a $\Delta E_{HOMO}$ of 0.17 eV showed a PCE of 7.3% with a $V_{oc}$ of 0.998 V and a $E_{loss}$ of ca. 0.60 eV (ref. 38). The device based on a 2D-conjugated D-A copolymer PBDB-T as donor and a low bandgap *n*-OS ITIC (ref. 20) ($E_g$ of 1.59 eV) as acceptor with a $\Delta E_{HOMO}$ of 0.18 eV demonstrated a PCE of 11.21% with a $V_{oc}$ of 0.899 V and a $E_{loss}$ of 0.64 eV (ref. 26). Notably, the lower $E_{loss}$ was attained with a relatively low $\Delta E_{HOMO}$, suggesting a low driving force needed for excitons dissociation in this system. These encouraging results provide a plausible approach to well remove the trade-off between the $V_{oc}$ and $J_{sc}$. Recently, we developed a series medium bandgap 2D-conjugated D-A copolymers based on bithienyl-benzodithiophene (BDTT) donor unit and fluorine-substituted benzotriazole (FBTA) acceptor unit[25,39], which showed good photovoltaic performance in the PSCs with the polymers as donor and n-OS as acceptors[11,25]. Among the polymers, J61 as donor blending with the *n*-OS ITIC as acceptor with a $\Delta E_{HOMO}$ of 0.16 eV in the non-fullerene PSCs displayed a high PCE of 9.53% with a $V_{oc}$ of 0.89 V

and a $E_{loss}$ of 0.68 eV (ref. 25). The high photovoltaic performance of the non-fullerene PSCs with the small $\Delta E_{HOMO}$ between donor and acceptor is a very important phenomenon for the development of high performance PSCs by increasing $V_{oc}$ and decreasing $E_{loss}$.

To probe the possibility of further increasing the $V_{oc}$ and PCE of the non-fullerene PSCs with smaller $\Delta E_{HOMO}$ value, herein we try to further decrease HOMO energy level of the BDTT-FBTA-based 2D-conjugated D-A copolymers by introducing trialkylsilyl substituents on BDTT unit. The trialkylsilyl substituents instead of alkyl side chains make HOMO energy level of the synthesized polymer **J71** down-shifted to −5.40 eV and its absorbance enhanced in comparison with its analogue polymer J52 with alkyl side chains[25]. Very interestingly, the non-fullerene PSC based on **J71** as donor and ITIC as acceptor with a small $\Delta E_{HOMO}$ of 0.11 eV demonstrates a high PCE of 11.41% with high $V_{oc}$ of 0.94 V, high $J_{sc}$ of 17.32 mA cm$^{-2}$ and a low $E_{loss}$ of 0.63–0.65 eV. The PCE of 11.41% is one of the highest efficiency of the single junction PSCs reported in literatures till now. The results indicate that the alkylsilyl substitution is an effective way in designing high performance conjugated polymer photovoltaic materials.

## Results

**Design and synthesis of J71.** In the molecular design of conjugated polymer donor materials for PSCs, the main chain engineering (such as alternative donor (D)-acceptor (A) copolymerization)[40,41] and side chain engineering (such as conjugated side chains and electron-withdrawing substituents)[30,42] are widely used strategy to tune absorption spectra and electronic energy levels of the conjugated polymers. Silicon (Si)-containing fused rings, such as dibenzosilole and dithienosilole, are representative donor units in the high performance D-A copolymer donor materials. Silicon bridging atom in the Si-containing fused rings plays an important role in improving the planarity of the backbone and modifying the electronic properties of the polymers[43–46]. The effect of the Si-atom on the electronic properties are mostly related to the bond interaction of the low-lying $\sigma^\star$ orbital of the Si atom with the $\pi^\star$ orbital of the aromatic units, which results in stabilization of the LUMO energy level and lowering of HOMO energy level[47]. For example, Si-bridged bithiophene (dithienosilole, DTS) has lower HOMO and LUMO energy levels compared with those of its C-bridged counterpart (CDT)[47]. In addition, the Si-bridging atom also showed a significant effect on the crystallinity of the polymer due to the longer C–Si bond length compared with the C–C bond, leading to stronger π–π stacking. Inspired by the special-function of the Si-atom connecting with conjugated system, herein we designed another BDTT-FBTA-based D-A copolymer **J71** by introducing alkylsilyl substituents on thiophene conjugated side chains of BDTT unit for down-shifting HOMO energy level and strengthening interchain interaction of the polymers by the $\sigma^\star$ (Si)–$\pi^\star$(C) bond interaction.

The polymer **J71** was synthesized by Stille-coupling polycondensation between **M1** (BDTT-Si) and **M2** (FTAZ-based monomer)[48], as shown in Fig. 1a. The structure of BDTT-Si obtained by X-ray crystallography was also shown in Fig. 1a, and the crystallographic data of BDTT-Si (CCDC number: 1478875) are listed in Supplementary Table 1 and Supplementary Table 2.

To better understand the effect of alkylsilyl substitution on the electronic properties of the monomer and polymer, we compared the absorption spectra and electronic energy levels of BDTT-Si with the alkylsilyl substituents and BDTT-C with alkyl substituents on the thiophene conjugated side chains of the BDTT units, as shown in Fig. 1b. It can be seen that the absorption maximum ($\lambda_{max}$) of BDTT-Si is at 380 nm which is

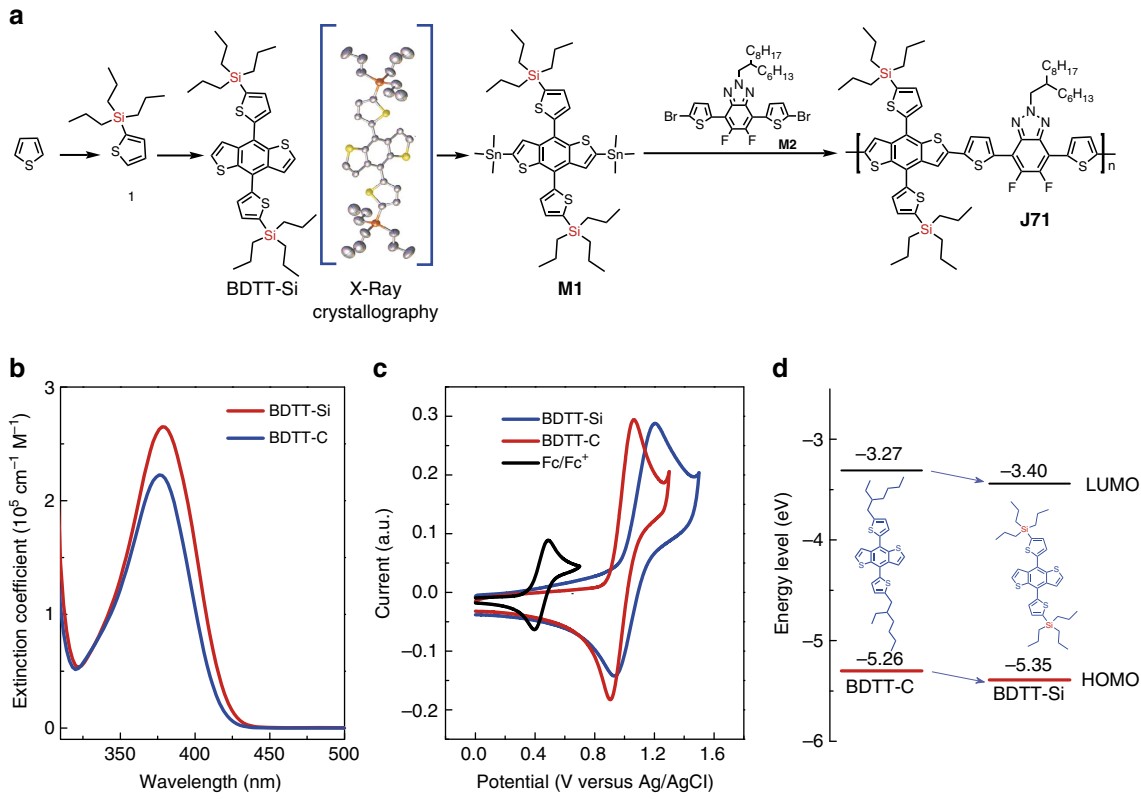

**Figure 1 | Synthetic route of J71 and the effect of the alkylsilyl substitution on the physicochemical properties of the monomers.** (**a**) Synthetic route of **J71** with the structure of BDTT-Si obtained by X-ray crystallography. (**b**) Absorption spectra of the monomers BDTT-Si and BDTT-C chloroform solutions with concentration of $1 \times 10^{-5}$ mol l$^{-1}$. (**c**) Cyclic voltammograms of BDTT-Si and BDTT-C in 0.1 mol l$^{-1}$ Bu$_4$NPF$_6$ acetonitrile solution at a scan rate of 20 mV s$^{-1}$, the ferrocene/ferrocenium (Fc/Fc$^+$) couple was also provided for an internal reference. (**d**) Energy level diagrams of BDTT-Si and BDTT-C.

5 nm red-shifted than that (375 nm) of BDTT-C. And the absorption coefficient of BDTT-Si is higher than that of BDTT-C, indicating that alkylsilyl substitution affords not only broader but also stronger absorption. HOMO energy levels of the two monomers were measured by cyclic voltammetry. The onset oxidation potentials ($\varphi_{ox}$) of BDTT-Si and BDTT-C are 0.99 and 0.90 V versus Ag/AgCl (Fig. 1c), and the HOMO energy levels ($E_{HOMO}$) were calculated to be $-5.35$ eV for BDTT-Si and $-5.26$ eV for BDTT-C (the calculation equation will be discussed below in the section on the electronic energy level measurements of **J71**). The LUMO levels of BDTT-Si and BDTT-C, estimated from their absorption edges and HOMO levels, are $-3.40$ and $-3.27$ eV respectively. Clearly, alkylsilyl substituted BDTT-Si has lower HOMO and LUMO enrgy levels compared with those of its alkyl substituted counterpart BDTT-C (Fig. 1d). The results suggest that the alkylsilyl side chain engineering can afford the similar effect of the Si–C bond interaction as that in Si-bridged aromatic compounds[47], while the alkylsilyl side chain approach developed in this work is more simple and convenient in down-shifting the HOMO energy level and strengthening the absorption. To further understand the effect of the alkylsilyl side chains on the electronic energy levels of the monomers, we performed the theoretical calculation by the DFT method on the molecules at the DFT B3LYP/6-31G(d) level with the Gaussian 03 program package. The calculated HOMO and LUMO energy levels of BDTT-Si are lower than those of BDTT-C (Supplementary Fig. 1), which is consistent with the experimental results mentioned above.

The synthesis details of **J71** were described in the Method section. **J71** exhibits good solubility in many chlorinated solvents, such as chloroform, chlorobenzene and dichlorobenzene. The number average molecular weight ($M_n$) of **J71** is 23.5 kDa with a polydispersity index of 2.0, which was measured by high temperature Gel permeation chromatography (GPC). Thermogravimetric analysis (TGA) demonstrated a good thermal stability of the polymer with a 5% weight-loss temperature at 354 °C, as shown in Supplementary Fig. 2.

**Absorption spectra and electronic energy levels.** Figure 2a shows the molecular structures of the polymer **J71** and the n-OS ITIC acceptor, and Fig. 2b displays the absorption spectra of **J71** solution and film together with the film absorpiton of ITIC for comparison. In solution, **J71** shows defined absorption profile with two peaks at about 528 and 573 nm, which can be ascribed to the vibronic bands associated with the (0-1) and (0-0) transitions respectively. The emergence of these features is believed to be associated with the partial aggregation of the fluorinated polymer chains. The absorption of **J71** film is red-shifted by about 14 nm along with a stronger (0-0) transition peak in the long wavelength range. The red-shift of its absorption in the solid film indicates more ordered structure and stronger $\pi$–$\pi$ stacking interaction, which should benefit higher hole mobility and better photovoltaic performance of the polymer. The absorption edge of **J71** film is at 632 nm, which correponds to an optical bandgap of 1.96 eV. The film maximum extinction coefficient of **J71** is $0.96 \times 10^5$ cm$^{-1}$, which is higher than its polymer analogue J52 with alkyl chain ($0.73 \times 10^5$ cm$^{-1}$)[25]. In addition, as one of the series of BDTT-FBTA-based copolymers[11,25,39], **J71** also demonstrates well complementary absorption with that of ITIC n-OS acceptor in the wavelength range of 400–800 nm, which is beneficial for light harvesting in the non-fullerene PSCs.

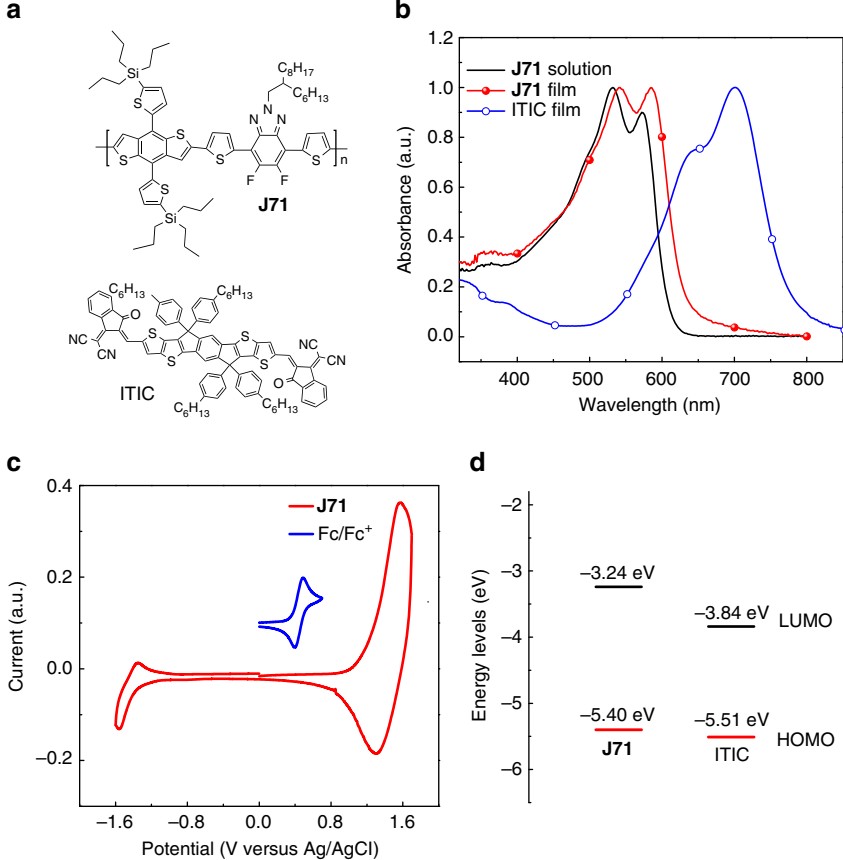

**Figure 2 | Chemical structure and physicochemical properties of J71.** (**a**) Chemical structures of **J71** polymer donor and ITIC n-OS acceptor.
(**b**) Absorption spectra of **J71** and ITIC. (**c**) Cyclic voltammogram of **J71** polymer film on a platinum electrode measured in 0.1 mol l$^{-1}$ Bu$_4$NPF$_6$ acetonitrile solutions at a scan rate of 20 mV s$^{-1}$, the inser figure (blue line) shows the Cyclic voltammogram of ferrocene/ferrocenium (Fc/Fc$^+$) couple used as an internal reference. (**d**) Energy level diagram of **J71** and ITIC.

The HOMO and LUMO energy levels of **J71** were measured by electrochemical cyclic voltammetry[49,50], for investigating the effect of the tripropylsilyl substitution on the electronic energy levels of the 2D-conjugated polymers. The HOMO/LUMO energy levels ($E_{HOMO}$/$E_{LUMO}$) can be calculated from onset oxidation/reduction potentials ($\varphi_{ox}$/$\varphi_{red}$) in the cyclic voltammograms according to the equations of $E_{HOMO}$/$E_{LUMO} = -e$ ($\varphi_{ox}$/$\varphi_{red} + 4.8 - \varphi_{Fc/Fc^+}$) (eV)[49,50] where $\varphi_{Fc/Fc^+}$ is the redox potential of ferrocene/ferrocenium (Fc/Fc$^+$) couple in the electrochemical measurement system, and the evergy level of Fc/Fc$^+$ was taken as 4.8 eV below vacuum. The electrochemical measurement was performed in a 0.1 M acetonitrile solution of tetrabutylammonium hexafluorophosphate (n-Bu$_4$NPF$_6$) with the sample (**J71** or ITIC) film deposited on Pt disc electrode as working electrode, Ag/AgCl as reference electrode. $\varphi_{Fc/Fc^+}$ was measured to be 0.44 V versus Ag/AgCl in this measurement system, and then the calculation equations are $E_{HOMO}$/$E_{LUMO} = -e$ ($\varphi_{ox}$/$\varphi_{red} + 4.36$) (eV).

Figure 2c shows the cyclic voltammegram of **J71** film, from which the onset oxidation potential ($\varphi_{ox}$) and onset reduction potential ($\varphi_{red}$) of **J71** were measured to be 1.04 and $-1.10$ V versus Ag/AgCl respectively (Supplementary Fig. 3). The HOMO energy level ($E_{HOMO}$) and LUMO energy level ($E_{LUMO}$) of **J71** were calculated to be $-5.40$ and $-3.24$ eV respectively (see Fig. 2d), according to the Equations mentioned above. Under the same experimental conditions, the $E_{HOMO}$ and $E_{LUMO}$ of ITIC were measured to be $-5.51$ and $-3.84$ eV respectively (see Fig. 2d and Supplementary Fig. 3b).

It should be noted that the $E_{HOMO}$ of **J71** is 0.19 and 0.08 eV deeper than that of its polymer analogue J52 with alkyl substituent ($E_{HOMO} = -5.21$ eV, see Supplementary Fig. 4) and J61 with alkylthio substituent ($E_{HOMO} = -5.32$ eV) respectively[25]. The results indicate that the alkylsilyl substitution (with $\sigma^\star$ (Si)–$\pi^\star$(C) bond interaction) is effective in lowering the HOMO energy level of the 2D-conjugated polymers. It should be mentioned that the electrochemical bandgap ($E_{LUMO}$ -$E_{HOMO}$) of **J71** is 2.16 eV, which is 0.2 eV larger than that (1.96 eV) of its optical bandgap. The larger electrochemical bandgap is a common phenomenon for the conjugated polymers and is reasonable in considering the energy barriers needed for the charge transfer in electrochemical oxidation and reduction reactions on the electrode.

**Photovoltaic properties**. PSCs were fabricated with a traditional device structure of ITO (indium tin oxide) /PEDOT: PSS (poly (3, 4-ethylenedioxythiophene): poly (styrene-sulfonate)) /**J71**: ITIC (1:1, w/w) /PDINO[51] (perylene diimide functionalized with amino N-oxide)/Al, where PDINO was chosen as the cathode interlayer for lowering the work funciton of Al[51]. The weight ratio of **J71** donor: ITIC acceptor is 1: 1 according to our recent work in the studies of non-fullerene PSCs based on J61:ITIC[25], considering the similar chemical structure of **J71** with J61. The active layers with a thickness of about 100 nm were prepared by spin-coating the **J71**: ITIC blend solution with a total blend concentration of 12 mg ml$^{-1}$ in chloroform at 3000 r.p.m.

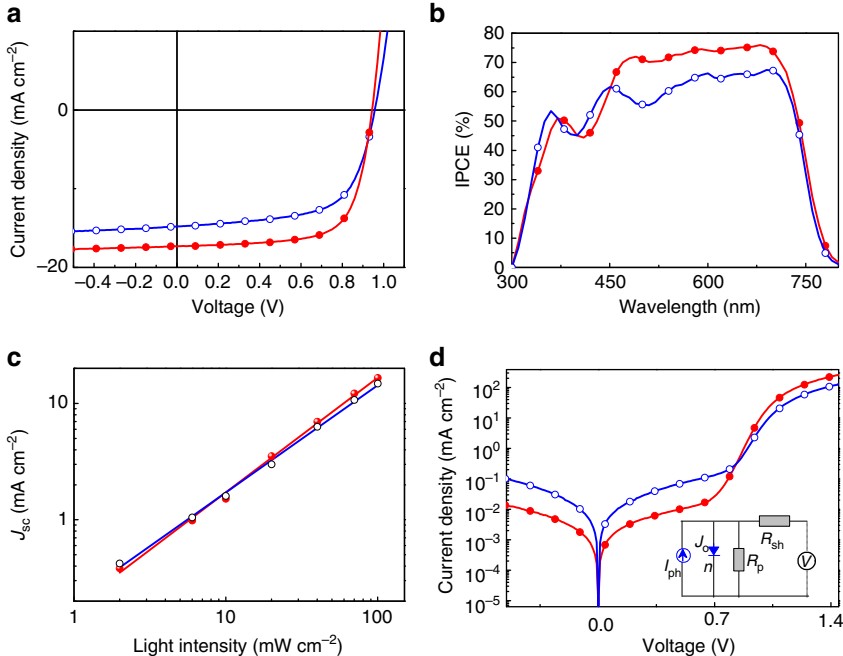

**Figure 3 | Photovoltaic performance of the PSCs based on J71:ITIC without (open circles) and with (filled circles) thermal annealing at 150 °C for 10 min.** (**a**) J–V curves of the champion PSCs, under the illumination of AM 1.5 G, 100 mW cm$^{-2}$, (**b**) IPCE spectra of the corresponding PSCs; (**c**) Light intensity dependence of $J_{sc}$ of the PSCs; (**d**) dark currents of the PSCs, the inset shows the equivalent circuit of the PSCs.

**Table 1 | Photovoltaic performance parameters of the PSCs based on J71:ITIC under the illumination of AM1.5 G, 100 mW cm$^{-2}$.**

| Devices | $V_{oc}$ (V) | $J_{sc}$ (mA cm$^{-2}$) | FF (%) | PCE (%) | $R_s$ $\Omega$ cm$^2$ | $R_{sh}$ k$\Omega$ cm$^2$ | $\mu_h$ $10^{-4}$ cm$^2$V$^{-1}$s$^{-1}$ | $\mu_e$ $10^{-4}$ cm$^2$V$^{-1}$s$^{-1}$ | $\mu_h/\mu_e$ |
|---|---|---|---|---|---|---|---|---|---|
| As-prepared | 0.96 0.96 ± 0.004* | 14.81 14.55 ± 0.56 | 63.63 63.85 ± 0.51 | 9.03 8.94 ± 0.24 | 10.38 | 0.57 | 2.08 | 0.96 | 2.17 |
| Thermal-annealed† | 0.94 0.94 ± 0.003 | 17.32 17.40 ± 0.37 | 69.77 68.09 ± 1.12 | 11.41 11.2 ± 0.29 | 1.13 | 1.9 | 3.78 | 3.07 | 1.23 |

*Average values with standard deviation were obtained from 30 devices.
†Thermal annealing at 150 °C for 10 min.

Figure 3a shows the current density–voltage (J–V) curves, and the corresponding incident photon to converted current efficiency (IPCE) spectra are shown in Fig. 3b. For a clear comparison, the detailed photovoltaic performance data are listed in Table 1. The as-prepared PSCs based on **J71**: ITIC showed a PCE of 9.03% with a high $V_{oc}$ of 0.96 V, a $J_{sc}$ of 14.81 mA cm$^{-2}$ and a FF of 63.63%. After thermal annealing at 150 °C for 10 min, the PCE of the PSC was further improved to 11.41% with a high $V_{oc}$ of 0.94 V, a high $J_{sc}$ of 17.32 mA cm$^{-2}$ and a FF of 69.77%. The PCE of 11.41% is one of the highest efficiency for the single junction PSCs reported in literatures so far. And very importantly, both high $V_{oc}$ and high $J_{sc}$ were achieved in the **J71**:ITIC-based PSCs, which is a big challenge for the high performance PSCs.

It can be seen that the PSCs based on **J71**: ITIC exhibited high $V_{oc}$ in the range of 0.94–0.96 V, which is certainly benefitted from the low-lying HOMO energy level of the alkylsilyl substituted **J71**. As mentioned in the Introduction part, a crucial issue in the studies of PSCs is to minimize the device photon energy loss ($E_{loss}$) which is defined as $E_{loss} = E_g - eV_{oc}$, where $E_g$ is the lowest optical band gap among the donor and acceptor components[33–35]. In the present PSCs based on **J71**: ITIC, the lowest $E_g$ is 1.59 eV for the ITIC acceptor with onset absorption at

782 nm (see Supplementary Fig. 5b). Therefore the $V_{oc}$ of 0.94–0.96 V results in a low $E_{loss}$ of 0.63–0.65 eV, which is smaller than that of most PSCs and approaching the empirically low threshold of 0.6 eV. It should be mentioned that using the onset absorption to determine $E_g$ is the commonly accepted and wide used method by different groups[33–35,52,53]. And this method can provide a straight forward comparison of our results with those results previously reported (see Supplementary Table 3). Recently, Scharber et al. proposed a more accurate method to measure the $E_g$ value of the active layer (blend of donor and acceptor) of the PSCs from IPCE spectrum[36]. With this method, we obtained a $E_g$ of 1.58 eV from the onset of IPCE spectrum of the PSC based on **J71**:ITIC (as shown in Supplementary Fig. 6) which is consistent with the $E_g$ value of 1.59 eV from the absorption edge of ITIC mentioned above.

The plot of $eV_{OC}$ against $E_g$, and the plots of PCE and IPCE$_{max}$ against $E_{loss}$ for the PSCs reported in literatures are provided in Supplementary Fig. 7, and the detailed data of the corresponding devices are listed in Supplementary Table 3 for a clear comparison. It should be noted that the PCE and the IPCE$_{max}$ (76.5%) of the PSC based on **J71**/ITIC are in fact the highest values among the fullerene and non-fullerene PSCs reported in literatures with $E_{loss}$ less than 0.65 eV. Such small $E_{loss}$ is

benefitted from the small $\Delta E_{HOMO}$ of 0.11 eV in the PSCs based on J71/ITIC.

The IPCE spectra of the PSCs based on J71: ITIC (Fig. 3b) demonstrate broad and high photo-response from 300 to 790 nm, which indicates high photo-conversion efficiency for the absorptions of both J71 polymer donor and ITIC acceptor. The IPCE spectra in the wavelength range of 650–790 nm are corresponding to the excitons dissociation of the ITIC acceptor with the hole transfer from the HOMO of ITIC to that of J71. The high IPCE values in this range confirm that efficient hole transfer process did occour even though the $\Delta E_{HOMO}$ between J71 donor and ITIC acceptor is only 0.11 eV. The $J_{sc}$ values integrated from the IPCE spectra are 16.564 mA cm$^{-2}$ for the thermal annealed device and 14.688 mA cm$^{-2}$ for the as-prepared device, which agrees well with those values obtained from the J–V curves within 5% mismatch (Table 1). It should be noted that the low $\Delta E_{HOMO}$ request can provide big chance in the molecular design of photovoltaic materials, such as for the polymer donor to further downshift its $E_{HOMO}$ toward a larger $V_{oc}$, thus it will be promising to address the big challenge of PSCs for maximizing $V_{oc}$ and $J_{sc}$ at the same time and ultimately getting a high-efficiency.

We also tried to fabricate the inverted PSCs with a device structure of ITO/ZnO /J71: ITIC (1:1, w/w) /MoO$_3$/Al. The inverted PSC with thermal annealing at 150 °C for 10 min showed a PCE of 10.7% with a $V_{oc}$ of 0.93 V, a $J_{sc}$ of 17.36 mA cm$^{-2}$ and a FF of 66.05%, as shown in Supplementary Fig. 8. The slightly lower PCE of the inverted device could be due to the electrode buffer layer materials used in the inverted PSCs, and the optimization of the inverted PSCs is underway.

The charge carrier mobilities of the PSCs were measured by space-charge-limited current (SCLC) method to investigate the effect of thermal annealing. The plots of the current density versus voltage of the devices for the mobility measurements are shown in Supplementary Fig. 9. For the as-cast blend film, their hole mobility $\mu_h$ and electron mobility $\mu_e$ are estimated to be $2.08 \times 10^{-4}$ cm$^2$ V$^{-1}$ s$^{-1}$ and $0.96 \times 10^{-4}$ cm$^2$ V$^{-1}$ s$^{-1}$ respectively with $\mu_h/\mu_e$ of 2.17, while after thermal annealing the $\mu_h$ and $\mu_e$ values were increased to $3.78 \times 10^{-4}$ cm$^2$ V$^{-1}$ s$^{-1}$ and $3.07 \times 10^{-4}$ cm$^2$ V$^{-1}$ s$^{-1}$ respectively with the improved $\mu_h/\mu_e$ ratio of 1.23. The improved photovoltaic performance of the PSCs with thermal annealing could be ascribed to the higher and more-balanced charge carrier mobilities of the thermal-treated J71: ITIC film. Figure 3c shows the dependance of $J_{sc}$ values on the light-intensity ($P_{light}$) which reflects the charge recombination behaviour of the devices. In general, the relationship between $J_{sc}$ and $P_{light}$ can be described as $J_{sc} \propto P_{light}^\alpha$. If the bimolecular recombination of the charge carriers is negligible, the power-law exponent $\alpha$ should be unity. The values of $\alpha$ for the as-prepared and thermal treated devices are 0.922 and 0.986, respectively (Fig. 3c). The higher $\alpha$ value of the thermal-treated device suggests the reduced charge recombination in its device, which correlates well with the balanced charge carrier mobilities of the devices mentioned above.

The effect of thermal annealing on the device performance was further studied by analysing the series resistance ($R_s$) and shunt resistance ($R_{sh}$) of the devices from their dark J–V curves (Fig. 3d), and the $R_s$ and $R_{sh}$ values are also listed in Table 1. The $R_s$ and $R_{sh}$ values of the as-prepared device are 10.38 $\Omega$ cm$^2$ and 0.57 k$\Omega$ cm$^2$ respectively, while after thermal annealing, $R_s$ was decreased to 1.13 $\Omega$ cm$^2$ and $R_{sh}$ was increased to 1.9 k$\Omega$ cm$^2$, suggesting the better overall diode characteristics after the thermal annealing, which is also reflected in the lower ideality factor $n$ (1.51 for the thermal-annealed device in comparison with 1.92 for the as-prepared device) and lowest dark saturation current density $J_0$ ($1.89 \times 10^{-10}$ mA cm$^{-2}$ for the

thermal-annealed device while it is $1.44 \times 10^{-8}$ mA cm$^{-2}$ for the as-prepared device).

**Morphological characterization**. For non-fullerene PSCs, morphology can be a determining factor governing the devcie performance[54,55]. Here, the microstructural features and surface morphologies of the neat J71 and ITIC films as well as their blend films with or wihtout thermal annealing were investigated by grazing incident wide-angle X-ray diffraction (GIWAXS) plots[56] and tapping-mode atomic force microscopy (AFM). Figure 4 shows the plots and images of the GIWAXS measurements. Strong diffraction peaks of the neat J71 film, as shown in Fig. 4b, reveal the semicrystalline structure and strong preference for face-on orientation in the polymer film. In addition, characteristic of the long range order and crystallinity in the film was observed in the in-plane direction. The high crystallinity is largely benefitted from its proper side chain[42] and fluorination effect[48]. For neat ITIC film, its lamellar (100) peak is located at 0.360 Å$^{-1}$ and $\pi$–$\pi$ stacking (010) peak is at 1.75 Å$^{-1}$ (Fig. 4a,e), corresponding to lamellar distance of 17.44 Å and a $\pi$–$\pi$ stacking distance of 3.58 Å. Its large azimuthal distribution of the diffraction peaks suggests randomly oriented crystallites, which is related to the steric effect of its tetrahexylphenyl substituents.

For the blend films, the GIWAXS plots demonstrated microstructural features with the diffraction patterns contributed from individual components (Fig. 4c,g). After thermal annealing, the J71: ITIC blend film is more prone to adopt a prominent face-on orientation (Fig. 4d,h) as evidenced by the small azimuthal distribution of the (010) $\pi$-$\pi$ stacking in the out-of-plane direction. Further looking into diffraction patterns, it can be found that, the peak intensities become significantly stronger accompanied by their narrower peak widths (Fig. 4d,h). Furthermore, a 3.85 Å $\pi$-$\pi$-stacking distance ($q = 1.63$ Å$^{-1}$) is seen in the thermal treated blend films, which is slighltly smaller than that of the as-cast J71: ITIC film (3.88 Å, $q = 1.62$ Å$^{-1}$). All the behaviour is related to the higher crystalline characteristics of the thermal treated blend film, and the combination of the preferred face-on orientation and the tight $\pi$-$\pi$-stacking of the polymer backbone is known to assist intermolecular charge transport and suppress the charge carrier recombination, which eventually improves the photovoltaic performance.

The AFM images, as shown in Supplementary Fig. 10, reveal that the blend films have relatively smooth surface with a root-mean-square (RMS) roughness of 0.722 nm for the as-cast J71: ITIC blend film and 0.741 nm for the thermal annealed blend film, indicating the good miscibility between the alkylsilyl substituted J71 polymer donor and ITIC acceptor. From the AFM phase images in Supplementary Fig. 10d, it can be seen that with the thermal annealing, enhanced crystalline domains with nano-networks around 20 nm are visualized. This highly inter-crystalline morphology asross the active layer enhances the polymer donor domain connectivity and thereby improves the hole transport, which correlates well with its superior device performance.

**Hole transfer and charge separation dynamics**. As mentioned above, it is likely that hole transfer from ITIC to J71 is highly efficient, despite the energy difference between HOMO levels ($\Delta E_{HOMO}$) of ITIC and J71 is only 0.11 eV which is much smaller than the empirical threshold of 0.3 eV for effective exciton dissociation. To confirm the assessment, we performed transient absorption spectroscopy measurement to investigate the charge transfer dynamics of photo-induced carriers in the blend film of J71: ITIC. The primary absorption peaks for J71 and ITIC are well separated in spectral domain (Fig. 2b), so we can extract the

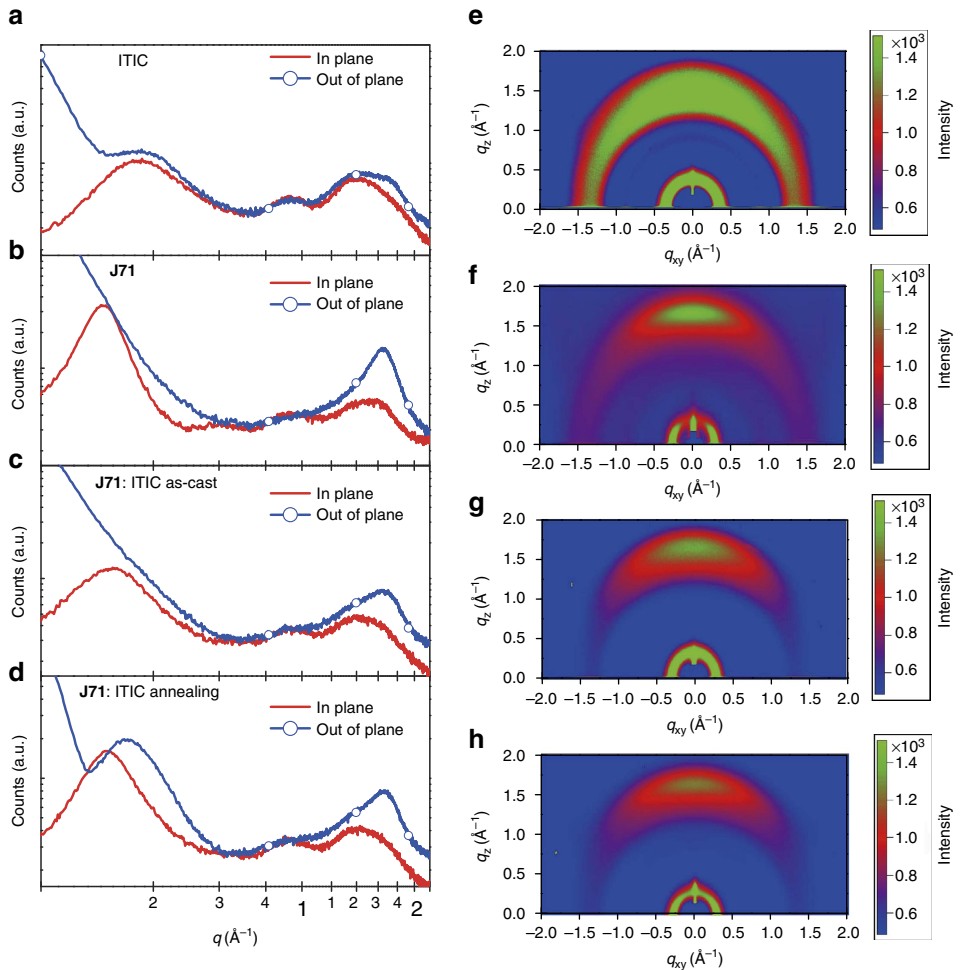

**Figure 4 | Plots and images of the GIWAXS measurements.** Line cuts of the GIWAXS images of (**a**) neat ITIC film and (**b**) neat **J71** film, (**c**) as cast **J71**: ITIC blend films and (**d**) thermal annealed **J71**: ITIC blend films. GIWAXS images of (**e**) the neat ITIC film, (**f**) neat **J71** film, (**g**) as acst **J71**: ITIC film and (**h**) thermal annealed **J71**: ITIC film.

spectral and temporal characteristics of hole transfer dynamics with selected excitation. The pump wavelengths of 540 and 710 nm were selected to excite **J71** and ITIC, respectively. The excitation density is kept in a weak regime below 1 μJ cm$^{-2}$ to avoid the effect of exciton-exciton annihilation. The pump-probe experiments measure the pump-induced differential change of the probe transmission, $\Delta T/T = (T_{\text{pump-on}} - T_{\text{pump-off}})/T_{\text{pump-off}}$.

Figure 5 shows the results of transient absorption spectroscopy with excitation wavelength at 710 nm. At this pump wavelength, only ITIC is excited as confirmed by the absence of transient absorption signal from the neat sample of **J71** (Supplementary Fig. 11). A bleaching signal peaked at 710 nm appears in both neat ITIC and the blend (Fig. 5a,b). This signal at the absorption peak of ITIC can be naturally ascribed to the ground state bleaching (GSB) of the transition in ITIC. In addition, clear bleaching peaks at 540 and 590 nm appear in the transient absorption spectrum of blend (Fig. 5b). These wavelengths are exactly the absorption peaks of **J71** (Fig. 2b). The spectral feature is also consistent with the GSB signal from **J71** observed with resonant excitation at 540 nm (Supplementary Fig. 11). Figure 5c compares the transient absorption spectra from the blend sample at different time delays with the initial GSB spectra observed in **J71** and ITIC under resonant excitations. Basically, the bleaching signals at 540 and 590 nm are built up with the decays of bleaching signal at 710 nm, suggesting the transfer of excitations

from ITIC to **J71**. The excitation photon energy (at 710 nm) is much smaller than that required for exciton absorption of the polymer, therefore, the bleaching signals (540 and 590 nm) cannot be ascribed to energy transfer process. Notably, the bleaching signals at about 540 nm and about 590 nm appear simultaneously with the decay of the GSB signals of ITIC at about 710 nm (Fig. 5c), such excitation transfer can be naturally assigned to the hole transfer since the LUMO level of ITIC is much lower than that of **J71**.

Figure 5d compares the dynamics probed at 710 nm for the films of neat ITIC and blend **J71**:ITIC in a normalized scale. For the early stage at less than 20 ps, the relaxation rate becomes dramatically faster in the blend film with respect to the neat ITIC, indicating the presence of additional relaxation channel of hole transfer in the blend film. We quantify the early-stage kinetics with a bi-exponetial decay function (Supplementary Fig. 12). The lifetime parameters for the two components are about 0.72 ps and about 15.0 ps respectively for the neat ITIC which decreases to about 0.29 ps and about 4.5 ps in the blends. Correspondingly, the buildup of signals probed at 540 and 590 nm in blend film is different from the abrupt rises observed in **J71** excited at 570 nm (Supplementary Fig. 11). The onset shows two exponential growth components with lifetime parameters of about 0.3 ps and about 4.8 ps, respectively (Fig. 5e), confirming the hole transfer is the primary origin of

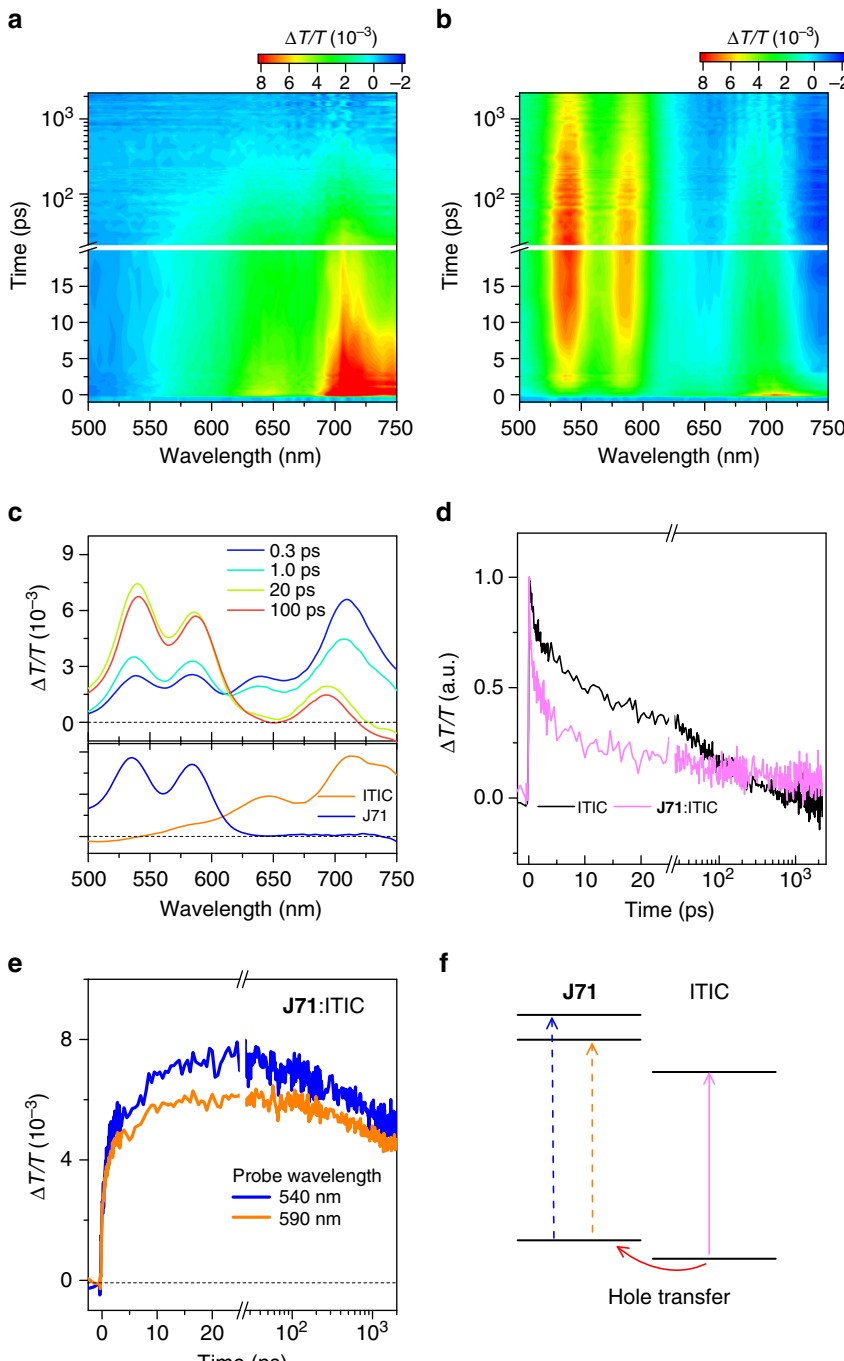

**Figure 5 | Transient absorption measurements for the study of hole transfer dynamics.** transient absorption signal recorded from the films of (**a**) neat ITIC and (**b**) **J71**: ITIC (1:1, w/w) blend excited by 710 nm. (**c**) Transient absorption spectra from the blend film exicted by 710 nm (orange line) at different time delays. The lower panel shows the transient absorption spectra recorded at 1 ps from ITIC excited by 710 nm and **J71** excited by 540 nm (blue line), respectively. (**d**) Dynamics probed at 710 nm recorded from the films of neat ITIC and blend **J71**: ITIC (1:1, w/w). (**e**) Dynamic curves probed at 540 and 590 nm recorded from the film of blend. (**f**) A schematic digram of the hole transfer in the film of **J71**: ITIC (1:1, w/w) blend.

the GSB signal at the absorption band of **J71**. With these results, we briefly summarize the dynamics in Fig. 5f where the hole transfer from ITIC is quite efficient with rate up to 3 ps$^{-1}$. Moreover, it is worthy noting that the GSB signals at 540, 590 and 710 nm in the blend film persist to nanosecond scale which is much longer than those in neat samples of **J71** and ITIC. The results suggest the existence of the long-lived dissociated excitons in the blend which is beneficial for electricity generation.

## Discussion

In conclusion, another BDTT-FBTA-based 2D-conjugated D-A copolymer **J71** with tripropylsilyl substituents on thiophene conjugated side-chain of BDTT unit, was synthesized. **J71** possesses a low-lying HOMO energy level at $-5.40$ eV and high film absorption extinction coefficients which is benefitted from the $\sigma^*$ (Si)$-\pi^*$(C) bond interaction with the trialkylsilyl substitution. Non-fullerene PSCs were fabricated with **J71** as donor and a

narrow bandgap n-OS ITIC as acceptor, and the PCE of the PSCs based on J71/ITIC (1:1, w/w) with thermal annealing at 150 °C for 10 min reached 11.41% with simultaneously high $V_{oc}$ of 0.94 V and a high $J_{sc}$ of 17.32 mA cm$^{-2}$. The PCE of 11.41% is one of the highest efficiencies of the single-junction PSCs reported in literatures till now. The high $V_{oc}$, which results in a low $E_{loss}$ of 0.63–0.65 eV for the PSC, is benefitted from the low-lying HOMO level of the J71 donor. The high $J_{sc}$ should be ascribed to the extraordinarily high exciton charge separation efficiency even though the HOMO energy difference between the donor and acceptor is quite small (only 0.11 eV). And the high hole transfer efficiency from ITIC to J71 was confirmed by transient absorption spectra. The results indicate that the alkylsilyl substitution is an effective way in designing high performance conjugated polymer photovoltaic materials and the driving force for the hole transfer from the acceptor to donor could be quite small in the non-fullerene PSCs with the n-OS as acceptor.

## Methods

**Materials and synthesis.** ITIC was purchased from Solarmer Materials Inc. The other chemicals and solvents were purchased from J&K, Alfa Aesar or TCI Chemical Co. The monomers and polymer J71 were synthesized according to Scheme 1. The Benzo[1,2-b:4,5-b']dithiophene-4,8-dione and M2 was synthesized according to the procedure reported in the literatures[39].

Monomer 1 was synthesized as follows: Under protection of argon, to a solution of thiophene (8.4 g, 100 mmol) in THF (100 ml) at −78 °C was added n-BuLi (40 ml, 2.5 M in hexane) slowly, the mixture was kept at −78 °C for 1 h and warmed slowly to room temperature. Then, chlorotripropylsilane (19.2 g, 100 mmol) was added, and the mixture stirred overnight. The mixture was extracted by diethyl ether twice, washed by water and brine. Further purification was carried out by column chromatography using hexane as eluent to obtain pure tripropyl(thiophen-2-yl)silane (1). as a colourless liquid. (22.3 g, 93% yield). [1]H NMR(400 MHz, CDCl$_3$), δ (p.p.m.): 7.57-7.56 (d, 1H), 7.22 (d, 1H), 7.16-7.15(t, 1H), 1.40-1.32 (m, 6H), 0.96-0.92(m, 9H), 0.80-0.75(m, 6H).

Monomer BDTT-Si was synthesized as follows: Under protection of argon, to a solution of compound 1 (7.2 g, 30 mmol) in THF (30 ml) at 0 °C was added n-BuLi (12 ml, 2.5 M in hexane), the mixture was kept at 0 °C for 15 min and heated to 50 °C and stirred for 2 h. Then, Benzo[1,2-b:4,5-b']dithiophene-4,8-dione (2.2 g, 10 mmol) was added quickly, and the mixture stirred for 4 h. After cooling down to the room temperature, SnCl$_2$·2H$_2$O (18 g, 80 mmol) in 10% HCl (35 ml) was added, and the mixture was stirred for 3 h. The mixture was extracted by diethyl ether twice, washed by water and brine. The crude product purified with column chromatography using petroleum ether as eluent to obtain pure 4,8-bis (5-(tripropylsilyl)thiophen-2-yl)benzo[1,2-b:4,5-b']dithiophene (BDTT-Si) as a light yellow solid. (2.8 g, 42% yield). [1]H NMR (400 MHz, CDCl$_3$), δ (p.p.m.): 7.65-7.64(d, 2H), 7.57-7.56 (d, 2H), 7.47-7.45 (d, 2H), 7.36-7.35(d, 2H), 1.53-1.42(m, 12H), 1.05-1.00(m, 18H), 0.92-0.86 (m, 12H). [13]C NMR (100 MHz, CDCl$_3$), δ (p.p.m.):145.0, 139.3, 139.1, 136.6, 134.9, 129.3, 127.7, 124.2, 123.6, 18.6, 17.6, 16.5.

Monomer M1 was synthesized as follows: To a solution of BDTT-Si (1.332 g, 2 mmol) in THF (20 ml) at −78 °C was added n-BuLi (2 ml, 2.5 M in hexane). After the addition, the mixture was kept at −78 °C for 40 min; trimethyltin chloride (6 ml, 1 M in THF) was added dropwise. The resulting mixture was stirred for 2 h at room temperature. Then, it was poured into water and extracted with diethyl ether, washed by water and brine and after drying over MgSO$_4$, the solvent was removed and the residue was recrystallized with methanol to afford a yellow crystal (5,5'-(2,6-bis(trimethylstannyl)benzo(1,2-b:4,5-b')dithiophene-4, 8-diyl)bis(thiophene-5,2-diyl))bis(tripropylsilane) (M1) (1.56 g, 78.8% yield). [1]H NMR (400 MHz, CDCl$_3$), δ (p.p.m.): 7.74-7.67(t, 2H), 7.60-7.59(d, 2H), 7.36(d, 2H), 1.53–1.45(m, 12H), 1.04–0.97(m, 18H), 0.91–0.87(m, 12H), 0.46-0.32(t, 18H). [13]C NMR (100 MHz, CDCl$_3$), δ (p.p.m.):205.9, 144.9, 142.3, 141.4, 137.8, 136.5, 133.9, 130.4, 128.0, 121.5, 30.0, 17.6, 16.6, 15.3, −7.4, −7.5, −9.3, −11.0, −11.1.

Polymer J71 was synthesized as follows: M1 (248 mg, 0.25 mmol) and M2 (175 mg, 0.25 mmol) and dry toluene (10 ml) were added to a 25 ml double-neck round-bottom flask. The reaction container was purged with argon for 20 min, and then Pd(PPh$_3$)$_4$ (10 mg) was added. After another flushing with argon for 20 min, the reactant was heated to reflux for 12 h. The reactant was cooled down to room temperature and poured into MeOH (200 ml), then filtered through a Soxhlet thimble, which was then subjected to Soxhlet extraction with methanol, hexane, and chloroform. The polymer J71 of 275 mg (Yield 91%) was recovered as solid from the chloroform fraction by precipitation from methanol, and was dried under vacuum. GPC: $M_n = 23.5$ kDa; $M_w/M_n = 2.0$. Anal. Calcd for C$_{66}$H$_{85}$F$_2$N$_3$S$_6$Si$_2$ (%): C, 65.68; H, 7.10; N, 3.48. Found (%):C, 64.79; H, 7.06; N, 3.55. [1]H NMR (CDCl$_3$, 400 MHz): δ (p.p.m.) 8.15–8.12 (br, 2H), 8.01–7.45 (br, 6H), 7.20–6.90 (br, 2H), 4.69 (br, 2H), 2.24–0.83 (br, 73H).

**General characterization.** [1]H NMR spectra were measured on a Bruker DMX–400 spectrometer with d–chloroform as the solvent and trimethylsilane as the internal reference. Ultraviolet–visible absorption spectra were measured on a Hitachi U–3010 Ultraviolet–vis spectrophotometer. Mass spectra were recorded on a Shimadzu spectrometer. Elemental analyses were carried out on a flash EA 1112 elemental analyser. Thermogravimetric analysis (TGA) was conducted on a Perkin–Elmer TGA–7 thermogravimetric analyser at a heating rate of 20 °C min$^{-1}$ under a nitrogen flow rate of 100 ml min$^{-1}$. Gel permeation chromatography (GPC) measurements was performed on Agilent PL-GPC 220 instrument with high temperature chromatograph, using 1,2,4-trichlorobenzene as the eluent at 160 °C. Photoluminescence (PL) spectra were measured with a Shimadzu RF-5301PC fluorescence spectrophotometer. Electrochemical cyclic voltammetry was performed on a Zahner IM6e Electrochemical Workstation under a nitrogen atmosphere. The cyclic voltammograms of J71 polymer film and ITIC film on a Pt disk electrode (working electrode) were measured with a potential scan rate of 20 mV s$^{-1}$ in an acetonitrile solution of 0.1 M tetrabutylammonium hexa-fluorophosphate (n-Bu$_4$NPF$_6$) with a Ag/AgCl reference electrode and a platinum wire counter electrode. The ferrocene/ferrocenium (Fc/Fc$^+$) couple was used as an internal reference. Thin films of J71 or ITIC were prepared by drop-casting 1.0 µl of their chloroform solutions (analytical reagent, 1 mg ml$^{-1}$) onto the working electrode and then dried in the air. The J71 or ITIC film and J71:ITIC blend film for morphology measurements were prepared by spin-coating the corresponding solution in chloroform with a concentration of 10 mg ml$^{-1}$ on indium tin oxide (ITO) glass at 3,000 rpm for 30 s. The film morphology was measured using an atomic force microscope (AFM, SPA-400) using the tapping mode.

Hole and electron mobilities were measured using the the space charge limited current (SCLC) method, with the hole-only device of ITO/PEDOT:PSS/J71: ITIC(1:1, w/w) /Au for hole mobility measurement and electron-only device of ITO/ZnO/J71: ITIC (1: 1, w/w) /PDINO/Al for electron mobility measurement. The SCLC mobilities were calculated by MOTT-Gurney equation:[57,58]

$$J = \frac{9\varepsilon_r\varepsilon_0\mu V^2}{8L^3} \tag{1}$$

Where J is the current density, $\varepsilon_r$ is the relative dielictric constant of active layer material usually 2–4 for organic semiconductor, herein we use a relative dielectric constant of 4, $\varepsilon_0$ is the permittivity of empty space, µ is the mobility of hole or electron and L is the thickness of the active layer, V is the internal voltage in the device, and $V = V_{app} - V_{bi}$, where $V_{app}$ is the voltage applied to the device, and $V_{bi}$ is the built-in voltage resulting from the relative work function difference between the two electrodes (in the hole-only and the electron-only devices, the $V_{bi}$ values are 0.2 and 0 V respectively).

Grazing-incidence wide-angle X-ray scattering (GIWAXS) measurements were conducted at PLS-II 9A USAXS beam line of the Pohang Accelerator Laboratory in Korea. X-rays coming from the in-vacuum undulator (IVU) were monchromated (wavelength $\lambda = 1.109\ 94$ Å) using a double crystal monochromator and focused both horizontally and vertically (450 (H) × 60 (V) µm$^2$ in fwhm at sample position) using K-B type mirrors. The GIWAXS sample stage was equipped with a 7-axis motorized stage for the fine alignment of sample, and the incidence angle of X-ray beam was set to be 0.13–0.135° for polymer films, ITIC film and their bend films. GIWAXS patterns were recorded with a 2D CCD detector (Rayonix SX165), and X-ray irradiation time was 6–9 s, dependent on the saturation level of the detector. Diffraction angles were calibrated using a sucrose standard (monoclinic, P21, $a = 10.8631$ Å, $b = 8.7044$ Å, $c = 7.7624$ Å, $\beta = 102.938°$), and the sample-to-detector distance was ∼231 mm.

For transient absorption spectroscopy, pump pulses with tunable wavelength were from an optical parametric amplifier pumped by a regenerative amplifier (Libra, Coherent, 1 kHz, 90 fs). The carrier dynamics was probed by a broadband supercontinuum light source that was generated by focusing a small portion of the beam of the amplifier onto a sapphire plate. The chirp of the probe supercontinuum was corrected with error to be less than 100 fs over the whole spectral range. The transient absorption signal was then analysed by a high speed charge-coupled device (S11071-1104, Hamamatsu) with a monochromater (Acton 2358, Princeton Instrument) at 1 kHz enabled by a custom-built board from Entwicklungsbuero Stresing. The angle between the polarizations of pump and probe beam was set at the magic angle.

**Device fabrication and characterization.** The PSCs were fabricated with a structure of ITO/PEDOT: PSS (40 nm)/active layer/PDINO/Al. A thin layer of PEDOT: PSS was deposited through spin-coating on precleaned ITO-coated glass from a PEDOT: PSS aqueous solution (Baytron P VP AI 4083 from H. C. Starck) at 2,000 rpm and dried subsequently at 150 °C for 15 min in air. Then the device was transferred to a nitrogen glove box, where the active blend layer of J71 and ITIC was spin-coated from its chloroform solution onto the PEDOT: PSS layer under a spin-coating rate of 3,000 r.p.m. After spin-coating, the active layers were annealed at 150 °C for 10 min for the devices with thermal annealing treatment. The thickness of the active layers is about 100 nm. Then methanol solution of PDINO at a concentration of 1.0 mg ml$^{-1}$ was deposited atop the active layer at 3,000 r.p.m. for 30 s to afford a PDINO cathode buffer layer with thickness of about 10 nm. Finally, top Al electrode was deposited in vacuum onto the cathode buffer layer at a pressure of about $5.0 \times 10^{-5}$ Pa. The active area of the devices was 4.7 mm$^2$. Optical microscope (Olympus BX51) was used to define the active area of

the devices. The current density–voltage ($J$–$V$) characteristics of the PSCs were measured in a nitrogen glovebox ($O_2 < 0.1$ p.p.m., $H_2O < 0.1$ p.p.m.) on a computer-controlled Keithley 2450 Source-Measure Unit. The voltage scan is from $-1.5$ to $1.5$ V with voltage step of 10 mV and delay time of 1 ms. Oriel Sol3A Class AAA Solar Simulator (model, Newport 94023A) with a 450W xenon lamp and an air mass (AM) 1.5 filter was used as the light source. The light intensity was calibrated to 100 mW cm$^{-2}$ by a Newport Oriel 91150V reference cell. Masks made by laser beam cutting technology with a well-defined area of 2.2 mm$^2$ were used to define the effective area for accurate measurement of the photovoltaic performance. All the measurements with mask or without mask gave consistent results with relative errors within 0.5%. Actually, the measurement with mask give a slightly higher PCE mainly due to its slightly higher FF. The PCE results in the manuscript are from the measurement without mask. PCE statistics were obtained using 30 individual devices fabricated under the same conditions. The input photon to converted current efficiency (IPCE) was measured by Solar Cell Spectral Response Measurement System QE-R3-011 (Enli Technology Co., Ltd., Taiwan). The light intensity at each wavelength was calibrated with a standard single-crystal Si photovoltaic cell.

**Data availability.** The data that support the findings of this study are available from the corresponding author on request.

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

## Acknowledgements

This work was supported by the Ministry of Science and Technology of China (973 project, no. 2014CB643501), NSFC (nos. 91433117, 91333204 and 21374124) and the Strategic Priority Research Program of the Chinese Academy of Sciences (no. XDB12030200). The authors would like to thank Dr Dan Deng and Prof. Zhixiang Wei for their help in measuring photovoltaic performance of the inverted structured PSCs.

## Author contributions

H.B., Z.-G.Z. and Y.L. designed the polymer J71, H.B. synthesized and characterized J71. L.G., Y.Y. and Z.-G.Z. carried out the device fabrication and characterization, L.X. provide the cathode buffer layer material. S.C. and C.Y. measured the GIWAXS diffraction patterns. Y.Z., C.Z. and M.X. measured TA spectra. Y.L. and Z.-G.Z. supervised the project and wrote the paper. The first two authors contributed equally to this work.

## Additional information

**Competing financial interests:** The authors declare no competing financial interests.

