## [Peer Review File · Nature Communications]

Reviewer #1 (Remarks to the Author):

Remarks to the authors:

The authors represent a 11.4% organic solar cell using a trialkylsilyl substituted 2D conjugated polymer J71 as donor and an n-type small molecule acceptor ITIC. As authors mentioned, n-type non-fullerene acceptors already reached 9-10% and they are very promising candidates to replace fullerene derivatives. The summary of the work from co-authors and the manuscript is organized well. I agree that this report demonstrates really high power conversion efficiencies for organic solar cells, however; it lacks scientific novelty and in depth understanding of the high efficiency achieved with the donor/acceptor blend which can be considered for Nature Communications.

Various derivations of the polymer J71 is already synthesized and published with the acceptor ITIC [1]. The authors argue that trialkylsilyl groups are added to lower the HOMO level and increase the Voc of the corresponding solar cells. Although there is 150 mV shift in HOMO levels, it is not reflected in Voc at all. Only 50 meV gain is observed in solar cells. Furthermore, measurement of the energetic levels is not clearly described at all. The polymer J71 shows a band gap of 2.46 eV calculated from Figure 2d. While optical absorption should yield completely different values.

Probably, one of the reason for this is that the energy levels are calculated from the solution CV measurements which is totally different than film behaviour. Using inconsistent values are kept cited in literature which build a huge pile of mis-lead information for the future publications. I would also argue that most of the high efficiency solar cells are reported in inverted structure which is more stable than conventional. The authors reported solar cells in conventional structure and no inverted architecture is provided.

More importantly, the Eloss section is discussed in a complete unscientific way. With polymer: fullerene solar cells, usually the Eloss is attributed from the polymer band gap to Voc. As non-fullerene acceptors absorb in the visible as well, some of the recent work related it to the lowest band gap to Voc. However, this should be clarified in the paper and the reasons should be mentioned. In addition, there is no report in literature that the HOMO-HOMO offset is a limitation for organic solar cells (authors do not provide any reference as well). The exciton binding energy is related to the LUMO-LUMO offset between the donor and acceptor molecule that facilitates exciton dissociation, which is usually >0.3 V (even it is stated that this value is empirical). [2] In this work, I measure a loss of 0.8 V, which is quite common in organic solar cells. In this case, $E_g - V_{oc}$ becomes 1V. In order to refer to the energetic losses and compare it to the other systems, sensitive measurements for absorption edges and energy levels should be performed. This section is the main motivation of the manuscript and there is no experimental proof of the arguments authors are demonstrating. Therefore, I would recommend sending it to another publishing group where sole device efficiency can be reported without further understanding.

[1] J. Am. Chem. Soc., 2016, 138, 4657–4664.

[2] J. Am. Chem. Soc., 2012, 134 (1), pp 685-692.

A. Summary of the key results: Well written

B. Originality of interest: Not novel (references and explanation is provided above)

C. Quality of data: Poor, reliable and detailed measurements for energy levels and energy losses should be performed.

D. Use of statistics: Good

E. Conclusions: Well written

F. Suggested improvements, experiments: Sensitive band gap measurements, inverted architecture devices

G. References: Should cover the recent non-fullerene work.

H. Clarity: Poor (reasons are provided in the above text)

Reviewer #2 (Remarks to the Author):

By using new design concept and clever synthetic chemistry, the authors of the manuscript presented a very interesting and convincing story that tributylsilyl side-chain substituted medium bandgap polymer (J71) can afford record high power conversion efficiency (PCE) of 11.4% with both high Voc of 0.94 V and high Jsc of 17.32 mA/cm² in non-fullerene PSCs. Besides the impressive high efficiency and the new chemistry used, the most interesting thing as disclosed by transient absorption spectroscopy on the photophysics in the new material system is that efficient hole transfer and long-lived exciton dissociation occurs under an extremely low ΔE_{HOMO} of 0.02 eV at the J71/ITIC interface. This unconventional feature ensures simultaneously obtaining the high VOC and high JSC toward record high efficiency in this work, and can provide new design concept in non-fullerene PSCs. The results are very interesting. Therefore, I think this is a nice work that can be published on Nature Communications after some minor revisions as indicated below:

1. In Figure 1c of the cyclic voltammogram, the internal reference (ferrocene/ferrocenium) and the scan rate should be given.
2. In Figure 1a (Page 6), the authors give the X-ray structure of BDTT-Si. The name of the structure should be labeled on the bottom of its structure. Also more information on the X-ray structure data should be given in the Supporting Information.
3. From the X-ray structure of BDTT-Si (Figure 1a), it can be seen that the two thiophene side-chains adopt a Trans-isomerism. However, DFT calculation on BDTT-Si adopts a Cis-isomerism (Figure S1), so as its counterpart (BDTT-C). Thus for convincing, DFT calculation on BDTT-C and BDTT-Si should be performed both on their Cis-isomerism, and the results should be compared with the experimental results.
4. In Figure 3c, the JSC data are missing, only leaving the fitting line curves. Please check and revised Figure 3c.
5. An interesting finding of this work is that efficient hole transfer can occur when the HOMO of donor is approaching that of acceptor. As ΔE_{HOMO} is an important consideration in new materials design. The significance of this finding on new material design should be discussed.

Reviewer #3 (Remarks to the Author):

Authors present an exciting study that achieves over 11% power conversion efficiency (PCE) of PSCs through the innovation on the donor polymer and non-fullerene acceptor blend. Instead of the common alkyl chains, the trialkylsilyl side chain is employed to develop wide bandgap D-A copolymer, which surprisingly not only lower the HOMO of the resultant polymer, but also enhanced polymer absorbance. This is indeed a new strategy for the functionalization of efficient polymers. The BHJ blend made with new polymer and known acceptor ITIC exhibited high power conversion efficiency (PCE) of 11.41% in PSC with Voc of 0.94 V and Jsc of 17.32 mA/cm². The further analyses of BHJ morphology and charge photogeneration dynamic were conducted through the assistance of GIWAXS and femtosecond transient absorption spectra. Overall, the whole body of study is incorporated with sufficient amount of experimental proofs and rational interpretation. The result is the state-of-art, accompanying the solid and deep analysis. This work will attract readership of society. Reviewer strongly recommend the publication of this work in Nature Communication after some minor concerns being addressed.

- 1) Authors insist a very small ΔE_{HOMO} as low as 0.02 eV between donor polymer and acceptor, but did not reveal the source for this data. This is actually a very sensitive value in the MS. Reviewer suggests that the Donor and Acceptor HOMO values should be taken under same condition for the

fair comparison.

2) In Figure 3c for the light intensity dependence of JSC, the liner line is the fitting curve of raw data plots. These plots need to be displayed in Fig. 3c for the better indication of original data.

3) In the Figure 5c, the hole transfer (after 710 nm excitation of ITIC) from ITIC to polymer donor would result two type charge polarons, ITIC radical anion and polymer radical cation, respectively. However, the rise-up signals (540 nm and 590 nm) match well with that polymer GSB signals, unlikely to be the polaron signals. Can the observed transient dynamic relate to the sensitization process (energy transfer) rather than hole transfer? Please comment on this.

Response to Reviewer #1:

1) Various derivations of the polymer J71 is already synthesized and published with the acceptor ITIC [1]. The authors argue that trialkylsilyl groups are added to lower the HOMO level and increase the Voc of the corresponding solar cells. Although there is 150 mV shift in HOMO levels, it is not reflected in Voc at all. Only 50 meV gain is observed in solar cells.

2) Furthermore, measurement of the energetic levels is not clearly described at all. The polymer J71 shows a band gap of 2.46 eV calculated from Figure 2d. While optical absorption should yield completely different values. Probably, one of the reason for this is that the energy levels are calculated from the solution CV measurements which is totally different than film behaviour. Using inconsistent values are kept cited in literature which build a huge pile of mis-lead information for the future publications.

Response: The two comments are related to the HOMO and LUMO energy levels of J71, and the electrochemical measurement of the electronic energy levels. Following the reviewer's comments and suggestion, we added a paragraph to clearly describe the electrochemical measurement method and cited 2 related literatures (Ref. 49, 50) in pp. 10-11:

“The HOMO and LUMO energy levels of J71 were measured by electrochemical cyclic voltammetry,^{49,50} for investigating the effect of the tributylsilyl substitution on the electronic energy levels of the 2D-conjugated polymers. The HOMO/LUMO energy levels (E_{HOMO}/E_{LUMO}) can be calculated from onset oxidation/reduction potentials (j_{ox}/j_{red}) in the cyclic voltammograms according to the equations of $E_{HOMO}/E_{LUMO} = -e (\overline{\phi} / \overline{\phi}_{red} + 4.8 - \square_{Fc/Fc^+})$ (eV)^{49,50} where $\overline{\phi}_{Fc^+}$ is the redox potential of ferrocene/ferrocenium (Fc/Fc^+) couple in the electrochemical measurement system, and the energy level of Fc/Fc^+ was taken as 4.8 eV below vacuum. The electrochemical measurement was performed in a 0.1 M acetonitrile solution of tetrabutylammonium hexafluorophosphate ($n\text{-Bu}_4\text{NPF}_6$) with the sample (J71 or ITIC) film deposited on Pt disc electrode as working electrode, Ag/AgCl as reference electrode. j_{Fc/Fc^+} was measured to be 0.44 V vs. Ag/AgCl in this measurement system, and then the calculation equations are $E_{HOMO}/E_{LUMO} = -e (\overline{\phi} / \overline{\phi}_{red} + 4.36)$ (eV).”

In addition, more detailed description of the electrochemical measurement conditions were reported in the Supporting Information in p. S2, lines 6-16.

We also re-measured the electrochemical cyclic voltammogram of J71 as shown in Figure 2(c) in the revised manuscript, re-checked the redox potential of Fc/Fc^+ which was used as an internal standard. After the careful calibration, now the HOMO level of J71 is corrected to be -5.40 eV which is 80 meV down-shifted in comparison with that (-5.32 eV) of J61 with alkylthio substituent. The Voc increase of ca. 50 mV for the PSCs with J71 as donor than that with J61 as donor is reasonable because Voc is not only related to the HOMO energy level of the donor, but also related to the charge recombination etc. in the devices.

In the revised manuscript, the LUMO level of J71 is corrected to be -3.24 eV. We added two sentences in pp.11-12 to explain the difference between the electrochemical bandgap and optical bandgap of J71: “It should be mentioned that the electrochemical bandgap ($E_{LUMO} - E_{HOMO}$) of J71 is 2.16 eV, which is 0.2 eV larger than that (1.96 eV) of its optical bandgap. The larger electrochemical bandgap is a common phenomenon for the conjugated polymers and is reasonable in considering the energy barriers needed for the charge transfer in electrochemical oxidation and reduction reactions on the electrode.”

In addition, we want to mention that the J71 film on working electrode was used in the electrochemical measurement. The electrochemical method for the measurements of HOMO and LUMO energy levels of conjugated polymers is originated from an early literature (Eckhardt, H., Shacklette, L.W., Jen, K.Y., Elsenbaumer, R.L., The electronic and electrochemical properties of poly(phenylene vinylene) and poly(thienylene vinylene): An experimental and theoretical study. *J. Chem. Phys.*, **91**, 1303-1315 (1989) (Ref. 49)), based on the following diagram:

Figure 1, reproduced with permission from AIP Publishing LLC. Original caption: Relationship between electrochemically measured onset potentials, $(E_o)_{ox}$ and $(E_o)_{red}$ and the ionization potential, IP, and band gap, E_g , derived from a simplified band structure of polymers.

And the method has been widely used in the determination of HOMO and LUMO energy levels of conjugated polymers in the field of conjugated polymer optoelectronic materials.

3) *I would also argue that most of the high efficiency solar cells are reported in inverted structure which is more stable than conventional. The authors reported solar cells in conventional structure and no inverted architecture is provided.*

Response: In conventional PSCs, low workfunction active metal such as Ca is commonly used as cathode which results in poorer stability. While in this study, we used an alcohol-soluble *n*-type organic semiconductor PDINO instead of Ca as cathode buffer layer in fabricating the conventional PSCs, and the poor stability problem of Ca is avoided. In addition, PDINO can't be used as the cathode buffer layer on ITO in the inverted PSCs due to its strong visible absorption. To follow the reviewer's suggestion, we tried to fabricate the inverted PSCs with a device structure of ITO/ZnO/J71: ITIC (1:1, w/w)/MoO₃/Al. The inverted PSC with thermal annealing at 150^oC for 10 min showed a PCE of 10.7% with a V_{oc} of 0.93 V, a J_{sc} of 17.36 mA/cm² and a FF of 66.05%, as shown in Figure S8 in SI. The slightly lower PCE of the inverted device could be due to the electrode buffer layer materials used in the inverted PSCs, and the optimization of the inverted PSCs is underway.

4) *More importantly, the E_{loss} section is discussed in a complete unscientific way. With polymer: fullerene solar cells, usually the E_{loss} is attributed from the polymer band gap to V_{OC} . As non- fullerene acceptors absorb in the visible as well, some of the recent work related it to the lowest band gap to V_{OC} .*

However, this should be clarified in the paper and the reasons should be mentioned.

Response: We added several sentences to more clearly describe the photon energy loss in p.14: “As mentioned in the Introduction part, a crucial issue in the studies of PSCs is to minimize the device photon energy loss (E_{loss}) which is defined as $E_{loss} = E_g - eV_{oc}$, where E_g is the lowest optical band gap among the donor and acceptor components.³³⁻³⁵ In the present PSCs based on **J71**: ITIC, the lowest E_g is 1.59 eV for the ITIC acceptor with onset absorption at 782 nm (see Figure S5(b) in SI). Therefore the V_{oc} of 0.94-0.96 V results in a low E_{loss} of 0.63~0.65 eV, ...” As for the E_g value selected for the calculation of E_{loss} , the E_g value of the narrow bandgap conjugated polymer donor was selected in the fullerene-based PSCs, because the E_g value of the polymer donor is lower than that of the fullerene acceptor. While for the non-fullerene PSCs with the narrow bandgap ITIC as acceptor and the broad bandgap conjugated polymer as donor, the E_g value of the acceptor ITIC should be selected in the calculation of the E_{loss} because it is the lowest E_g value among the donor and acceptor.

5) *In addition, there is no report in literature that the HOMO-HOMO offset is a limitation for organic solar cells (authors do not provide any reference as well). The exciton binding energy is related to the LUMO-LUMO offset between the donor and acceptor molecule that facilitates exciton dissociation, which is usually >0.3 V (even it is stated that this value is empirical).[2]*

Response: Actually, in any type of PSCs, both donor and acceptor materials absorb photons to produce excitons, and the excitons will diffuse to the donor/acceptor interface where the excitons will dissociate into electrons in the LUMO of the acceptor and holes in the HOMO of the donor. For the conventional polymer donor-fullerene (such as PCBM) acceptor system, both conjugated polymer donor and PCBM acceptor in the active layer absorb photons to produce excitons. The excitons will diffuse towards the interface of polymer/PCBM where the electrons of the excitons in the polymer donor phase will transfer to the LUMO level of the PCBM acceptor and the holes of the excitons in the PCBM acceptor phase will transfer to the HOMO of the polymer donor to form electrons in the acceptor phase and holes in the donor phase. (see p.724 in *Acc. Chem. Res.*, 2012, 45, 723–733). The driving force for the electron transfer from the polymer donor to PCBM acceptor is the LUMO-LUMO offset, while the driving force for the hole transfer from PCBM acceptor to polymer donor is the HOMO-HOMO offset of the donor and acceptor. But because the HOMO level of PCBM is very low (ca. -6.0 eV), and the HOMO level of the conjugated polymer donors is commonly higher than -5.5 eV, the HOMO-HOMO offset is always larger than 0.5 eV which is high enough for the high efficiency hole transfer from PCBM acceptor to the polymer donor, therefore the researchers only concern the LUMO-LUMO offset in the PSCs with fullerene as acceptor. However, for the non-fullerene PSCs, the HOMO level of the organic acceptor is around -5.5~-5.6 eV, so that the HOMO-HOMO offset should also be considered. In addition, in order to give more reference literatures on the charge transfer processes, we revised some sentences in p.3 in the “Introduction” part: “In PSCs, due to the low dielectric permittivity and relatively large excitons binding energy of organic semiconductor, polymer donor and acceptor with cascading energy levels (the LUMO and HOMO levels of the conjugated polymer donor should be higher than the corresponding LUMO and HOMO levels of the acceptor) are specially required to provide a driving force for excitons dissociation at the heterojunction interface.²⁷⁻³⁰ ...”

6) In this work, I measure a loss of 0.8 V, which is quite common in organic solar cells. In this case, $E_g - V_{oc}$ becomes 1V. In order to refer to the energetic losses and compare it to the other systems, sensitive measurements for absorption edges and energy levels should be performed. This section is the main motivation of the manuscript and there is no experimental proof of the arguments authors are demonstrating. Therefore, I would recommend sending it to another publishing group where sole device efficiency can be reported without further understanding.

Response: We carefully checked the absorption edges for determining the optical E_g . Figure S5 in SI shows the absorption spectra and fluorescence spectra of J71 and ITIC films. And we added several sentences to explain the E_g measurement in the middle of p. 14: “It should be mentioned that using the onset absorption to determine E_g is the commonly accepted and wide used method by different groups.^{33-35,52,53}. And this method can provide a straight forward comparison of our results with those results previously reported (see Table S3 in SI). Recently, Scharber *et al.* proposed a more accurate method to measure the E_g value of the active layer (blend of donor and acceptor) of the PSCs from IPCE spectrum.³⁶ With this method, we obtained a E_g of 1.58 eV from the onset of IPCE spectrum of the PSC based on J71:ITIC (as shown in Figure S6 in SI) which is consistent with the E_g value of 1.59 eV from the absorption edge of ITIC mentioned above.”

The energy levels of polymer donor and ITIC acceptor were also re-measured under the same measurement conditions, as shown in Figure S5 in SI. The details of the measurements were described in pp. 10-11. (also see our Response to the comments (1) and (2) above)

After the careful determination of the E_g and electronic energy levels, we revised the description on the photon energy loss in p. 14: “As mentioned in the Introduction part, a crucial issue in the studies of PSCs is to minimize the device photon energy loss (E_{loss}) which is defined as $E_{loss} = E_g - eV_{oc}$, where E_g is the lowest optical band gap among the donor and acceptor components.³³⁻³⁵ In the present PSCs based on J71: ITIC, the lowest E_g is 1.59 eV for the ITIC acceptor with onset absorption at 782 nm (see Figure S5(b) in SI). Therefore the V_{oc} of 0.94-0.96 V results in a low E_{loss} of 0.63~0.65 eV, which is smaller than that of most PSCs and approaching the empirically low threshold of 0.6 eV.”

We think our results are very important and the results have been clearly described. Therefore, this manuscript is suitable for the publication in *Nature Commun.*

Response to Reviewer #2:

1) In Figure 1c of the cyclic voltammogram, the internal reference (ferrocene/ferrocenium) and the scan rate should be given.

Response: The internal reference was given in Figure 1c and the scan rate (at a scan rate of 20 mV s^{-1}) was provided in the caption of Figure 1c.

2) In Figure 1a (Page 6), the authors give the X-ray structure of BDTT-Si. The name of the structure should be labeled on the bottom of its structure. Also more information on the X-ray structure data should be given in the Supporting Information.

Response: The name of structure of BDTT-Si obtained from the X-ray is labeled as “X-ray crystallography” in Figure 1a and the caption of Figure 1(a) was revised to “Synthetic route of J71 with the structure of BDTT-Si obtained by X-ray crystallography”. We added a sentence in the last paragraph of

p.6 to give the X-ray structure data: “The structure of BDTT-Si obtained by X-ray crystallography was also shown in Figure 1a, and the crystallographic data of BDTT-Si (CCDC number: 1478875) were listed in Table S1 and Table S2 in the supporting information (SI).”

3) *From the X-ray structure of BDTT-Si (Figure 1a), it can be seen that the two thiophene side-chains adopt a Trans-isomerism. However, DFT calculations on BDTT-Si adopt a Cis-isomerism (Figure S1), so as its counterpart (BDTT-C). Thus for convincing, DFT calculations on BDTT-C and BDTT-Si should be performed both on their Cis-isomerism, and the results should be compared with the experimental results.*

Response: There is an error in drawing the molecular structures (cis-structure) in the original Figure S1(b). Actually, we carried out the DFT calculations with a trans-structure of BDTT-C and BDTT-Si. Now we corrected the molecular structures to trans-structures in Figure S1(b). And we reported and compared the calculation results with the experimental results in p.8: “In order to further understand the effect of the alkylsilyl side chains on the electronic energy levels of the monomers, we performed the theoretical calculation by the DFT method on the molecules. The calculated HOMO and LUMO energy levels of BDTT-Si are lower than those of BDTT-C (Figure S1 in SI) which is consistent with the experimental results mentioned above.”

4) *In Figure 3c, the JSC data are missing, only leaving the fitting line curves. Please check and revised Figure 3c.*

Response: Thanks to the reviewer for pointing out the problem! Now we provided the J_{sc} data points in Figure 3c in the revised manuscript in Page 13.

5) *An interesting finding of this work is that efficient hole transfer can occur when the HOMO of donor is approaching that of acceptor. As ΔE_{HOMO} is an important consideration in new materials design. The significance of this finding on new material design should be discussed.*

Response: Following the reviewer’s suggestion, we added a sentence in middle of p.15 to discuss the importance of the finding of low ΔE_{HOMO} : “It should be noted that the low ΔE_{HOMO} request can provide big chance in the molecular design of photovoltaic materials, such as for the polymer donor to further downshift its E_{HOMO} toward a larger V_{oc} , thus it will be promising to address the big challenge of PSCs for maximizing V_{oc} and J_{sc} at the same time and ultimately getting a high efficiency.”

Response to Reviewer #3:

1) *Authors insist a very small ΔE_{HOMO} as low as 0.02 eV between donor polymer and acceptor, but did not reveal the source for this data. This is actually a very sensitive value in the MS. Reviewer suggests that the Donor and Acceptor HOMO values should be taken under same condition for the fair comparison.*

Response: Following reviewer’s suggestion, the energy levels of J71 polymer donor and ITIC acceptor were re-measured in our laboratory under the same measurement conditions, as shown in Figure S5 in SI. The details of the measurements were described in pp. 10-11. (also see our response to the comments (1) and (2) of Reviewer#1) Now, the HOMO and LUMO energy levels of **J71** are -5.40 and -

3.24 eV respectively. We measured the HOMO and LUMO energy levels of ITIC under the same experimental conditions and reported the results in p. 11: “Under the same experimental conditions, the E_{HOMO} and E_{LUMO} of ITIC were measured to be -5.51 and -3.84 eV respectively (Figure S3b in SI and Figure 2d).” Consequently, the ΔE_{HOMO} becomes 0.11 eV which is still much lower than the empirical requirement of larger than 0.3 eV, and the energy levels and the ΔE_{HOMO} values were updated in the revised manuscript.

2) *In Figure 3c for the light intensity dependence of J_{SC} , the linear line is the fitting curve of raw data plots. These plots need to be displayed in Fig. 3c for the better indication of original data.*

Response: The J_{sc} data points were provided in Fig. 3c in the revised manuscript.

3) *In the Figure 5c, the hole transfer (after 710 nm excitation of ITIC) from ITIC to polymer donor would result two type charge polarons, ITIC radical anion and polymer radical cation, respectively. However, the rise-up signals (540 nm and 590 nm) match well with that polymer GSB signals, unlikely to be the polaron signals. Can the observed transient dynamic relate to the sensitization process (energy transfer) rather than hole transfer? Please comment on this.*

Response: We appreciate the reviewer’s insightful comment. Indeed, the photo-induced bleaching signals (540 nm and 590 nm) are irrelevant to the polaron signals but match well with the GSB signals. Since the excitation photon energy (at 710 nm) is much smaller than the exciton absorption of the polymer, the bleaching signals (540 nm and 590 nm) cannot be ascribed to energy transfer process. Notably, the onset of bleaching signals (540 nm and 590 nm) appear simultaneously with the enhanced decay of the GSB signals of ITIC (Figure 5b), which strongly supports that the bleaching signals (540 nm and 590 nm) is induced by hole transfer from ITIC to polymer. According to the reviewer’s suggestion, we discussed the results in p.21: “The excitation photon energy (at 710 nm) is much smaller than that required for exciton absorption of the polymer, therefore, the bleaching signals (540 nm and 590 nm) cannot be ascribed to energy transfer process. Notably, the bleaching signals at ca. 540 nm and ca. 590 nm appear simultaneously with the decay of the GSB signals of ITIC at ca. 710 nm (Figure 5c), such excitation transfer can be naturally assigned to the hole transfer since the LUMO level of ITIC is much lower than that of **J71**.”

Reviewer #1 (Remarks to the Author)

The authors revised the manuscript according to the comments and added related parts into the manuscript. The manuscript can be accepted in its current form for publication in Nature Communications.

- A. Good
- B.High
- C.Strong
- D.Good
- E.Good
- F.Have been amended
- G.Appropriate
- H.Good

Reviewer #2 (Remarks to the Author)

The authors have supplemented experiments and revised the manuscript according to the reviewers' comments and revision suggestions. Now the revised manuscript can be accepted for publication at its present form.

Reviewer #3 (Remarks to the Author)

The comments from reviewer have been well addressed in this version. Therefore, reviewer would like to suggest publishing this MS in Nature Communications without changes.